# Rare Chromone Derivatives from the Marine-Derived *Penicillium citrinum* with Anti-Cancer and Anti-Inflammatory Activities

**DOI:** 10.3390/md19010025

**Published:** 2021-01-08

**Authors:** Yi-Cheng Chu, Chun-Hao Chang, Hsiang-Ruei Liao, Ming-Jen Cheng, Ming-Der Wu, Shu-Ling Fu, Jih-Jung Chen

**Affiliations:** 1Institute of Traditional Medicine, School of Medicine, National Yang-Ming University, Taipei 112, Taiwan; xbox88888@ym.edu.tw; 2Institute of Biopharmaceutical Sciences, Pharmaceutical Sciences, National Yang-Ming University, Taipei 112, Taiwan; howard860212@ym.edu.tw; 3Graduate Institute of Natural Products, College of Medicine, Chang Gung University, Taoyuan 333, Taiwan; liaoch@mail.cgu.edu.tw; 4Bioresource Collection and Research Center (BCRC), Food Industry Research and Development Institute (FIRDI), Hsinchu 300, Taiwan; cmj@firdi.org.tw (M.-J.C.); wmd@firdi.org.tw (M.-D.W.); 5Faculty of Pharmacy, School of Pharmaceutical Sciences, National Yang-Ming University, Taipei 112, Taiwan; 6Department of Medical Research, China Medical University Hospital, China Medical University, Taichung 404, Taiwan

**Keywords:** *Penicillium citrinum*, chromone derivatives, anti-inflammatory activity, anti-cancer activity

## Abstract

Three new and rare chromone derivatives, epiremisporine C (**1**), epiremisporine D (**2**), and epiremisporine E (**3**), were isolated from marine-derived *Penicillium citrinum*, together with four known compounds, epiremisporine B (**4**), penicitrinone A (**5**), 8-hydroxy-1-methoxycarbonyl-6-methylxanthone (**6**), and isoconiochaetone C (**7**). Among the isolated compounds, compounds **2**–**5** significantly decreased fMLP-induced superoxide anion generation by human neutrophils, with IC_50_ values of 6.39 ± 0.40, 8.28 ± 0.29, 3.62 ± 0.61, and 2.67 ± 0.10 μM, respectively. Compounds **3** and **4** exhibited cytotoxic activities with IC_50_ values of 43.82 ± 6.33 and 32.29 ± 4.83 μM, respectively, against non-small lung cancer cell (A549), and Western blot assay confirmed that compounds **3** and **4** markedly induced apoptosis of A549 cells, through Bcl-2, Bax, and caspase 3 *signaling* cascades.

## 1. Introduction

There are many natural products isolated from marine-derived fungi. These compounds are important source of biologically effective secondary metabolites, which are very interesting and important for drug discovery. In particular, a large number of natural products with biological activities are found in the genus *Penicillium*, which has diverse biological activities such as antibacterial, antifungal, antitumor, and antiviral activities [1,2,3,4,5,6,7,8,9,10,11,12,13,14,15,16,17,18]. Diverse dihydroisocoumarins [4], citrinin [15], benzopyrans [17], benzophenones [18] and their derivatives were isolated from *Penicillium citrinum* in the past studies. Many of these isolated compounds showed anti-bacterial [4,17,18], anti-fungal [15], and anti-cancer [18] activities.

Human neutrophils are known to play a critical role in the pathogenesis of various inflammatory diseases [19,20]. In response to different stimuli, activated neutrophils secrete a series of cytotoxins, such as superoxide anion (O_2_^•–^), granule proteases, and bioactive lipids [19,21,22]. Suppression of inappropriate activation of neutrophils by drugs was proposed as a way to combat inflammatory diseases [23].

According to statistics from Taiwan’s Ministry of Health and Welfare, cancer remained the top killer in Taiwan for many years [24]. The apoptosis-related proteins, such as Bcl-2, Bax*,* and caspase-3, regulate cancer cell apoptosis or survival, which was confirmed to be related to many cancers and diseases [25,26,27]. In a preliminary screening, the methanolic extract of *P. citrinum* showed anti-inflammatory and anti-cancer activities in vitro. The current chemical investigation of this fungus led to the isolation of three new chromone derivatives, epiremisporine C (**1**), epiremisporine D (**2**), and epiremisporine E (**3**), along with four known compounds. The structural elucidation of **1**–**3** and anti-inflammatory and anti-cancer properties of **1–7** are described herein.

## 2. Results and Discussion

### 2.1. Fermentation, Extraction, and Isolation

In this study, the marine-derived fungal strain *Penicillium citrinum* (BCRC 09F0458) was cultured in solid-state culturing conditions, in order to enrich the diversity of the fungal secondary metabolites. Chromatographic isolation and purification of the *n*-BuOH-soluble fraction of an EtOH extract of *Penicillium citrinum* on a silica gel column and preparative thin-layer chromatography (TLC) obtained three new (**1**–**3**) and four known compounds (**4**–**7**) (Figure 1).

### 2.2. Structural Elucidation

Compound **1** was isolated as a yellowish amorphous powder. Its molecular formula, C_31_H_26_O_12_, was determined on the basis of the positive HR-ESI-MS ion at *m*/*z* 613.13200 [M + Na]^+^ (calcd. 613.13219) and supported by the ^1^H- and ^13^C- NMR data. The IR spectrum showed the presence of hydroxyl (3428 cm^−1^), ester carbonyl (1744 cm^−1^), and conjugated carbonyl (1655 cm^−1^) groups. The ^1^H- and ^13^C-NMR data of **1** showed the presence of two hydroxy groups, two methyl groups, three methoxy groups, two pairs of meta-coupling aromatic protons, two methylene protons, and three methine protons. The signals at δ 12.11 and 12.30 exhibited two chelated hydroxyl groups with the carbonyl group. Comparison of the ^1^H and ^13^C NMR data of **1** with those of epiremisporine B [16] suggested that their structures were closely related, except that the 2′-methoxyl group of **1** replaced the 2′-hydroxy group of epiremisporine B [16]. This was supported by both HMBC correlations between OMe-2′ (δ_H_ 3.50) and C-2′ (δ_C_ 111.1) and ROESY correlations between OMe-2′ (δ_H_ 3.50) and H-3′ (δ_H_ 2.86). The relative configuration of **1** was elucidated on the basis of ROESY experiments. The ROESY cross-peaks between H-3/H-4, H-3/H-3′, H-3/H_α_-4′, OMe-2′/H-3′, and H-3/H-16 suggested that H-3, H-4, H-3′, OMe-2′, and COOMe-2 are α-oriented, and COOMe-2′ is β-oriented. To further confirm the relative configuration of **1**, a computer-assisted 3D structure was obtained by using the molecular-modeling program CS CHEM 3D Ultra 16.0, with MM2 force-field calculations for energy minimization. The calculated distances between H-3/H-4 (2.210 Å), H-3/H-3′ (2.491 Å), OMe-2′/H-3′ (2.305 Å), and H-3/H-16 (2.371 Å) were all less than 4 Å (Figure 2). This was consistent with the well-defined ROESY observed for each of these H-atom pairs. The absolute configuration of **1** was evidenced by the CD Cotton effects at 332.5 (Δ*ε* +8.81), 293.0 (Δ*ε* −1.23), 258.5 (Δ*ε* +15.60), 239.5 (Δ*ε* −4.12), and 206.5 (Δ*ε* +4.09) nm, in analogy with those of epiremisporine B [16]. The ^1^H- and ^13^C-NMR resonances were fully assigned by the ^1^H–^1^H COSY, HSQC, ROESY, and HMBC experiments (Figure 3). On the basis of the above data, the structure of **1** was elucidated, as shown in Figure 1, and named epiremisporine C.

Compound **2** was obtained as an amorphous powder. The ESI–MS demonstrated the quasi-molecular ion [M + Na]^+^ at *m*/*z* 627, implying a molecular formula of C_32_H_28_O_12_, which was confirmed by the HR-ESI-MS (*m*/*z* 627.12902 [M + Na]^+^, calcd. 627.14784) and by the ^1^H- and ^13^C-NMR data. The IR spectrum showed the presence of hydroxyl (3480 cm^−1^), ester carbonyl (1763 cm^−1^), and conjugated carbonyl (1657 cm^−1^) groups. The signal at δ 12.36 exhibited a chelated hydroxyl group with the carbonyl group. Comparison of the ^1^H and ^13^C NMR data of **2** with those of epiremisporine C (**1**), suggested that their structures were closely related, except that the 11-methoxyl group of **2** replaced the 11-hydroxy group of **1**. This was supported by both HMBC correlations between OMe-11 (δ_H_ 3.91) and C-11 (δ_C_ 160.0) and ROESY correlations between OMe-11 (δ_H_ 3.91) and H-10 (δ_H_ 6.58). The relative configuration of **2** was elucidated on the basis of ROESY experiments. The ROESY cross-peaks between H-3/H-4, H-3/H-3′, H-3/H_α_-4′, OMe-2′/H-3′, and H-3/H-16 suggested that H-3, H-4, H-3′, OMe-2′, and COOMe-2 were α-oriented, and COOMe-2′ was β-oriented. To further confirm the relative configuration of **2**, a computer-assisted 3D structure was obtained by using the molecular-modeling program CS CHEM 3D Ultra 16.0, with MM2 force-field calculations for energy minimization. The calculated distances between H-3/H-4 (2.200 Å), H-3/H-3′ (2.484 Å), OMe-2′/H-3′ (2.306 Å), and H-3/H-16 (2.329 Å) were all less than 4 Å (Figure 4). This was consistent with the well-defined ROESY observed for each of these H-atom pairs. Compound **2** showed similar CD Cotton effects [330.5 (Δ*ε* +5.39), 290.5 (Δ*ε* –6.24), 262.5 (Δ*ε* +19.72), 238.5 (Δ*ε* –2.06), and 207.0 (Δ*ε* +13.72) nm] compared with **1** and epiremisporine B [16]. Thus, **2** possessed a 2*S*,3*R*,2′*R*,3′*S*-configuration. On the basis of the above data, the structure of **2** was elucidated, as shown in Figure 1, and named epiremisporine D, which was further confirmed by the ^1^H-^1^H COSY, ROESY (Figure 5a), DEPT, HSQC, and HMBC (Figure 5b) experiments.

Compound **3** was isolated as an amorphous powder. The ESI–MS demonstrated the quasi-molecular ion [M + Na]^+^ at *m*/*z* 627, implying a molecular formula of C_32_H_28_O_12_, which was confirmed by the HR–ESI–MS (*m*/*z* 627.12919 [M + Na]^+^, calcd. 627.14784) and by the ^1^H- and ^13^C-NMR data. The IR spectrum showed the presence of hydroxyl (3466 cm^−1^), ester carbonyl (1761 and 1740 cm^−1^), and the conjugated carbonyl (1657 cm^−1^) groups. The signal at δ 12.50 exhibited a chelated hydroxyl group with the carbonyl group. Comparison of the ^1^H and ^13^C NMR data of **3** with those of epiremisporine D (**2**) suggested that their structures were closely related, except that the 2′β-methoxyl group of **3** replaced the 2′α-methoxyl group of **2**. This was supported by both HMBC correlations between OMe-2′ (δ_H_ 3.11) and C-2′ (δ_C_ 107.6), and the ROESY correlations between OMe-2′ (δ_H_ 3.11) and H_β_-4′ (δ_H_ 2.85). The relative configuration of **3** was elucidated on the basis of ROESY experiments. The ROESY cross-peaks between H-3/H-4, H-3/H-3′, H-3/H_α_-4′, OMe-2′/H-4′, and H-3/H-16 suggested that H-3, H-4, H-3′, COOMe-2, and COOMe-2′ were α-oriented, and OMe-2′ was β-oriented. To further confirm the relative configuration of **3**, a computer-assisted 3D structure was obtained by using the molecular-modeling program CS CHEM 3D Ultra 16.0, with MM2 force-field calculations for energy minimization. The calculated distances between H-3/H-4 (2.169 Å), H-3/H-16 (2.285Å), H-3/H-3′ (2.445 Å), and OMe-2′/H_β_-4′ (3.682Å) were all less than 4 Å (Figure 6). This was consistent with the well-defined ROESY observed for each of these H-atom pairs. Compound **3** showed similar CD Cotton effects [331.0 (Δ*ε* +4.01), 286.5 (Δ*ε* −7.51), 261.5 (Δ*ε* +19.77), 230.5 (Δ*ε* −4.98), and 207.5 (Δ*ε* +12.68) nm], compared to the literature data [16]. Thus, **3** possessed a 2*S*,3*R*,2′*S*,3′*S*-configuration. The ^1^H- and ^13^C-NMR resonances were fully assigned by ^1^H–^1^H COSY, ROESY (Figure 7a), HSQC, and HMBC (Figure 7b) experiments. On the basis of the above data, the structure of **3** was elucidated, as shown in Figure 1, and named epiremisporine E.

The correlations between the dihedral angles (H3′-C3′-C4′-H4′α and H3′-C3′-C4′-H4′β) and the vicinal coupling constants (*J*_3′, 4′α_ and *J*_3′, 4′β_) of compounds **1**–**3** and related analogues [16] are summarized in Table 1. The dihedral angles were calculated by using the molecular-modeling program CS CHEM 3D Ultra 16.0, with the MM2 force-field calculations for energy minimization. The correlations between dihedral angles (H3′-C3′-C4′-H4′α and H3′-C3′-C4′-H4′β) and vicinal coupling constants (*J*_3′, 4′α_ and *J*_3′, 4′β_) of compounds **1**–**3** were consistent with the Karplus relationship. The 2′*S,*3′*S*-configuration slightly decreased the *J*_3′, 4′β_ value from 11.3~12.8 to 8.3~10.3 compared to the 2′*R,*3′*S*-configuration. These data could also support the structural confirmation of the new compounds **1**–**3**.

### 2.3. Structure Identification of the Known Isolated Compounds

The known isolated compounds were readily identified by a comparison of physical and spectroscopic data (UV, IR, ^1^H-NMR, [α]_D_, and MS) with corresponding authentic samples or literature values. They included epiremisporine B (**4**) [16] (Appendix A; Appendix A), penicitrinone A (**5**) [15,28] (Appendix A), 8-hydroxy-1-methoxycarbonyl-6-methylxanthone (**6**) [10] (Appendix A), and isoconiochaetone C (**7**) [16] (Appendix A).

### 2.4. Biological Studies

#### 2.4.1. Inhibitory Activities on Neutrophil Pro-Inflammatory Responses

The anti-inflammatory effects of the isolated compounds from *Penicillium citrinum* were evaluated by their ability to suppress formyl-L-methionyl-L-leucyl-L-phenylalanine (fMLP)-induced O_2_^•–^ generation by human neutrophils. The anti-inflammatory activity data are shown in Table 2. The clinically used anti-inflammatory agent, ibuprofen, was used as the positive control. From the results of our anti-inflammatory tests, epiremisporine D (**2**), epiremisporine E (**3**), epiremisporine B (**4**), and penicitrinone A (**5**) exhibited inhibition (IC_50_ values ≤ 8.28 μM) of superoxide anion generation by human neutrophils, in response to fMLP. Thus, our study suggests *Penicillium citrinum* and its isolated compounds (especially **2**, **3**, **4**, and **5**) could be further developed as potential candidates for the treatment or prevention of various inflammatory diseases.

#### 2.4.2. Cytotoxic Effects and Selectivity of Compounds **1**–**7**

In this study, the cytotoxic activities of seven compounds (**1**–**7**) against A549 (human lung carcinoma) and HT-29 (human colon carcinoma) cells were studied; shown in Table 3. Among the isolated compounds, compounds **3**, **4**, and **5** exhibited potent cytotoxic activities with IC_50_ values of 43.82 ± 6.33, 32.29 ± 4.83, and 49.15 ± 6.47 μM against A549 cells, respectively. In addition, compound **4** exhibited cytotoxic activities with an IC_50_ value of 50.88 ± 2.29 μM against HT-29 cells. According to the data in Table 2, compounds **3** and **5** showed selective cytotoxicity against A549 cancer cells. Among the chromone derivatives (**1**–**4**), epiremisporine B (**4**) (with 2′-hydroxy group) exhibited a more effective cytotoxic activity than its analogue, epiremisporines C–E (**1**–**3**) (without 2′-hydroxy group) against the A549 and HT-29 cells. New compound, epiremisporines E (**3**) (with 2′β-methoxyl group) exhibited a stronger anticancer activity than its analogues, epiremisporines C and D (**1** and **2**) (with 2′α-methoxyl group) against A549 cells.

#### 2.4.3. New Compound 3 Inhibited Proliferation of A549 Cells

The known compounds, epiremisporine B (**4**) [16] and penicitrinone A (**5**) [28], were reported to exhibit anticancer activities in previous studies. Epiremisporine E (**3**) was selectively tested for clonogenic assay as it is a new compound and possesses cytotoxic activity against A549. The effect of compound **3** on colony formation of A549 cells was examined using the clonogenic assay (Figure 8). The A549 cell colonies were visualized as blue discs, through crystal violet staining. It was clearly observed that compound **3** (25 μM) significantly reduced the colony formation of A549 cells. Moreover, compound **3** almost completely inhibited the colony formation at 50 μM.

#### 2.4.4. Effects of Epiremisporine E (**3**) and Epiremisporine B (**4**) on Protein Expressions of Pro-caspase 3 and Cleaved-caspase 3 in A549 Cells

Caspase 3 activation is a hallmark of apoptosis. Caspase 3 activation involves the cleavage of pro-caspase 3 (the inactive precursor form of caspase 3), leading to the formation of cleaved-caspase 3 (which is the active caspase 3). Upon apoptosis, the pro-caspase 3 would decrease and the cleaved-caspase 3 would increase accordingly. We further investigated whether epiremisporine E (**3**) and epiremisporine B (**4**) were able to influence these enzymatic activities of caspase 3. The results showed that compounds **3** and **4** suppressed pro-caspase 3 and increased the cleaved-caspase 3 (Figure 9 and Figure 10). Furthermore, compounds **3** and **4** markedly induced apoptosis of A549 cells through caspase 3-dependent pathways.

#### 2.4.5. Effects of Compounds **3** and **4** on Protein Expressions of Bax and Bcl-2 in A549 Cells

To determine whether compounds **3** and **4** could influence the expression of proteins related to A549 cells apoptosis, compounds **3** and **4** (6.25, 12.5, and 25 μM) were added to A549 cells. Figure 9 and Figure 10 showed that the expression level of pro-apoptotic protein, bax was obviously higher with 25 μM treatment of compound **3** or **4** than with 12.5 or 6.25 μM treatment. On the contrary, the cells treated with 25 μM of compound **3** or **4** showed higher Bcl-2 (anti-apoptotic protein) expression than that treated with 12.5 or 6.25 μM. The results showed that compounds **3** and **4** suppressed the expression of Bcl-2 and increased bax expression.

## 3. Experimental Section

### 3.1. General Procedures

Optical rotations were measured using a Jasco *P*-2000 polarimeter (Japan Spectroscopic Corporation, Tokyo, Japan) in CHCl_3_. Circular dichroism (CD) spectra were obtained on a J-715 spectropolarimeter (Jasco, Easton, MD, USA). Ultraviolet (UV) spectra were obtained on a Hitachi U-2800 Double Beam Spectrophotometer (Hitachi High-Technologies Corporation, Tokyo, Japan). Infrared (IR) spectra (neat or KBr) were recorded on a Shimadzu IRAffinity-1S FT-IR Spectrophotometer (Shimadzu Corporation, Kyoto, Japan). Nuclear magnetic resonance (NMR) spectra, including correlation spectroscopy (COSY), rotating frame nuclear Overhauser effect spectroscopy (ROESY), heteronuclear multiple-bond correlation (HMBC), and heteronuclear single-quantum coherence (HSQC) experiments, were acquired using a BRUKER AVIII-500 or a BRUKER AVIII-600 spectrometer (Bruker, Bremen, Germany), operating at 500 or 600 MHz (^1^H) and 125 or 150 MHz (^13^C), respectively, with chemical shifts given in the ppm (δ), using CDCl_3_ as an internal standard (peak at 7.263 ppm in ^1^H NMR and 77.0 ppm in ^13^C NMR spectrum). Electrospray ionization (ESI) and high-resolution electrospray ionization (HRESI)-mass spectra were recorded on a Bruker APEX II Mass Spectrometer (Bruker, Bremen, Germany). Silica gel [70–230 mesh (63–200 μm) and 230–400 mesh (40–63 μm), Merck] was used for column chromatography (CC). Silica gel 60 F-254 (Merck, Darmstadt, Germany) was used for thin-layer chromatography (TLC) and preparative thin-layer chromatography (PTLC).

### 3.2. Fungal Material

The fungal strain *Penicillium citrinum* BCRC 09F458 was isolated from waste water, which was collected from Hazailiao, Dongshi, Chiayi, Taiwan, in 2009. The fungal strain was identified as *Penicillium citrinum* (family Trichocomaceae) by the BCRC center, based on cultural and anamorphic data. The rDNA-ITS (internal transcribed spacer) region was used for further identification. After searching the GenBank database through BLAST (nucleotide sequence comparison program), it was found to have a 100% similarity to *P. citrinum*. *P. citrinum* BCRC 09F458 was stored in the Biological Resources Collection and Research Center (BCRC) of the Food Industry Research and Development Institute (FIRDI).

#### Cultivation and Preparation of the Fungal Strain

We kept *P. citrinum* BCRC 09F0458 on potato dextrose agar (PDA), and cultivated the strain on PDA at 25 °C for one week, and finally harvested the spores with sterile water. The spores (5 × 10^5^) were seeded into 300 mL shake flasks containing 50 mL RGY medium (3% rice starch, 7% glycerol, 1.1% polypeptone, 3% soybean powder, 0.1% MgSO_4_, and 0.2% NaNO_3_), and cultivated with shaking (150 rpm) at 25 °C, for 3 days. After the mycelium enrichment step, an inoculum mixing 100 mL mycelium broth and 100 mL RGY medium was inoculated into plastic boxes (25 cm × 30 cm) containing 1.5 kg sterile rice and cultivated at 25 °C for producing rice, and the above mentioned RGY medium was added for maintaining the growth. After 21 days of cultivation, the rice was harvested, and used as a sample for further extraction.

### 3.3. Extraction and Isolation

The rice of the *P. citrinum* BCRC 09F0458 (1.5 kg) was extracted with 95% EtOH (3 × 10 L, 3 d each) at room temperature. The ethanol extract was concentrated under reduced pressure, and was partitioned with *n*-BuOH/H_2_O (1:1, v/v) to afford *n*-BuOH soluble fraction (36.2 g), H_2_O soluble fraction (13.0 g), and insoluble fraction (500 mg). The *n*-BuOH fraction (fraction A, 36.2 g) was purified by column chromatography (CC) (1.6 kg of silica gel, 70–230 mesh ((63–200 μm); *n*-hexane/EtOAc 25:1–0:1, 1500 mL) to afford 13 fractions: A1–A13. Fraction A2 (1.54 g) was subjected to medium pressure liquid chromatography (MPLC) (69 g of silica gel, 230–400 mesh (40–63 μm)); *n*-hexane/acetone 1:0–0:1, 700 mL-fractions) to give 11 subfractions: A2-1–A2-11. Fraction A2-6 (126 mg) was further purified by semipreparative normal-phase high performance liquid chromatography (HPLC) (silica gel; *n*-hexane/dichloromethane/EtOAc, 6:3:1) to obtain 8-hydroxy-1-methoxycarbonyl-6-methylxanthone (**6**) (14.9 mg). Fraction A6 (1.18 g) was subjected to MPLC (53 g of silica gel, 230–400 mesh (40–63 μm); *n*-hexane/acetone 1:0–0:1, 500 mL-fractions) to give 13 subfractions: A6-1–A6-13. Fraction A6-9 (150 mg) was further purified by preparative TLC (silica gel; *n*-hexane/EtOAc, 1:9) to afford penicitrinone A (**5**) (18.3 mg). Fraction A9 (1.44 g) was subjected to MPLC (65 g of silica gel, 230–400 mesh (40–63 μm)); dichloromethane/EtOAc 1:0–2:3, 650 mL-fractions) to give 12 subfractions: A9-1–A9-12. Fraction A9-6 (128 mg) was further purified by preparative TLC (silica gel; *n*-hexane/acetone, 3:2) to afford epiremisporine C (**1**) (12.2 mg). Fraction A10 (0.98 g) was subjected to MPLC (44 g of silica gel, 230–400 mesh (40–63 μm); *n*-hexane/acetone 1:0–0:1, 450 mL-fractions) to give 10 subfractions: A10-1–A10-10. Fraction A10-4 (120 mg) was further purified by preparative TLC (silica gel; *n*-hexane/EtOAc, 7:3) to afford isoconiochaetone C (**7**) (11.2 mg). Fraction A11 (2.38 g) was subjected to MPLC (107 g of silica gel, 230–400 mesh (40–63 μm); *n*-hexane/acetone 1:0–0:1, 1000 mL-fractions) to give 14 subfractions: A11-1–A11-14. Fraction A11-2 (138 mg) was further purified by semipreparative normal-phase HPLC (silica gel; *n*-hexane/EtOAc, 1:1) to afford epiremisporine D (**2**) (14.6 mg). Fraction A11-4 (168 mg) was further purified by preparative TLC (silica gel; dichloromethane/methanol, 19:1) to afford epiremisporine B (**4**) (23.8 mg). Fraction A12 (1.28 g) was subjected to MPLC (58 g of silica gel, 230–400 mesh (40–63 μm); *n*-hexane/EtOAc 1:0–0:1, 600 mL-fractions) to give 10 subfractions: A12-1–A12-10. Fraction A12-1 (122 mg) was further purified by preparative TLC (silica gel; *n*-hexane/dichloromethane/acetone, 5:3:2) to afford epiremisporine E (**3**) (13.7 mg).

Epiremisporine C (**1**): [α${]}_{D}^{25}$ = +527.6° (*c* 0.22, CHCl_3_); UV (MeOH) λ_max_ nm (log ε): 241 (4.35), 326 (3.50) nm; ^1^H NMR data, see Table 4; ^13^C NMR data, see Table 5.

Epiremisporine D (**2**): [α${]}_{D}^{25}$ = +526.8° (*c* 0.10, CHCl_3_); UV (MeOH) λ_max_ nm (log ε): 237 (4.25), 317 (3.52) nm; ^1^H NMR data, see Table 4; ^13^C NMR data, see Table 5.

Epiremisporine E (**3**): [α${]}_{D}^{25}$ = +561.3° (*c* 0.09, CHCl_3_); UV (MeOH) λ_max_ nm (log ε): 235 (4.30), 315 (3.54) nm; ^1^H NMR data, see Table 4; ^13^C NMR data, see Table 5.

Epiremisporine B (**4**): Yellow amorphous powder; [α${]}_{D}^{25}$ = +523.6° (*c* 0.16, CHCl_3_); ^1^H NMR data, see Appendix A; ^13^C NMR data, see Appendix A; HRESIMS: *m*/*z* 599.11558 [M + Na]^+^ (calcd. for C_30_H_24_O_12_ + Na, 599.11654).

Penicitrinone A (**5**): Orange crystalline powder; [α${]}_{D}^{25}$ = +102.6° (*c* 0.12, MeOH); ^1^H NMR (600 MHz, CDCl_3_) δ 1.32 (3H, d, *J* = 7.2 Hz, Me-4), 1.33 (3H, d, *J* = 7.0 Hz, Me-3′), 1.42 (3H, d, *J* = 6.4 Hz, Me-2′), 1.44 (3H, d, *J* = 6.7 Hz, Me-3), 2.11 (3H, s, Me-5), 2.21 (3H, s, Me-4′), 3.12 (1H, qd, *J* = 7.2, 0.9 Hz, H-4), 3.16 (1H, qd, *J* = 7.0, 4.1 Hz, H-3′), 4.61 (1H, qd, *J* = 6.4, 4.1 Hz, H-2′), 4.96 (1H, qd, *J* = 6.7, 0.9 Hz, H-3), 6.37 (1H, s, H-7), 8.33 (1H, br s, OH-5′); HRESIMS: *m*/*z* 381.17094 [M + H]^+^ (calcd. for C_23_H_24_O_5_ + H, 381.17020).

8-Hydroxy-1-methoxycarbonyl-6-methylxanthone (**6**): Yellow amorphous solid; ^1^H NMR (600 MHz, CDCl_3_) δ 2.44 (3H, s, Me-6), 4.03 (3H, s, COOMe-1), 6.64 (1H, br d, *J* = 1.4 Hz, H-5), 6.76 (1H, br d, *J* = 1.4 Hz, H-7), 7.31 (1H, dd, *J* = 7.3, 1.1 Hz, H-2), 7.53 (1H, dd, *J* = 8.5, 1.1 Hz, H-4), 7.74 (1H, dd, *J* = 8.5, 7.3 Hz, H-3), 12.15 (1H, s, OH-8); HRESIMS: *m*/*z* 285.07730 [M + H]^+^ (calcd. for C_16_H_12_O_5_ + H, 285.07630).

Isoconiochaetone C (**7**): Colorless needles (MeOH), m.p. 99–100.5 °C; [α${]}_{D}^{25}$ = +79.3° (*c* 0.18, MeOH); ^1^H NMR (600 MHz, CDCl_3_) δ 2.15 (1H, dddd, *J* = 14.0, 8.5, 2.5, 1.4 Hz, H_β_-2), 2.32 (1H, dddd, *J* = 14.0, 9.4, 7.4, 6.8 Hz, H_α_-2), 2.40 (3H, s, Me-8), 2.77 (1H, ddd, *J* = 18.0, 9.4, 2.5 Hz, H_β_-3), 3.17 (1H, dddd, *J* = 18.0, 8.5, 7.4, 1.4 Hz, H_α_-3), 3.50 (3H, s, OMe-1), 4.94 (1H, dt, *J* = 6.8, 1.4 Hz, H-1), 6.63 (1H, br s, H-9), 6.71 (1H, br s, H-7), 12.55 (1H, s, OH-10); HRESIMS: *m*/*z* 247.09688 [M + H]^+^ (calcd. for C_14_H_14_O_4_ + H, 247.09703).

### 3.4. Biological Assay

The anti-inflammatory effects of the isolated compounds from *Penicillium citrinum* were evaluated by suppressing fMLP-induced O_2_^•–^ generation by human neutrophils. In addition, anti-cancer activity was evaluated by cytotoxicity assay and Western blot analysis.

#### 3.4.1. Preparation of Human Neutrophils

Human neutrophils from the venous blood [21] of healthy, adult volunteers (20–35 years old) were isolated using a standard method of dextran sedimentation, prior to centrifugation in a Ficoll Hypaque gradient and hypotonic lysis of erythrocytes, as previously described [29]. Purified neutrophils containing >98% viable cells, as determined by the trypan blue exclusion method, were resuspended in HBSS buffer at pH 7.4 and were maintained at 4 °C, prior to use [30].

#### 3.4.2. Measurement of O_2_^•–^ Generation

The assay for measurement of O_2_^•–^ generation was based on the SOD-inhibitable reduction of ferricytochrome c [31]. In brief, neutrophils (1 × 10^6^ cells/mL) pretreated with the various test agents at 37 °C for 5 min were stimulated with fMLP (1 μmol/L) in the presence of ferricytochrome c (0.5 mg/mL). Extracellular O_2_^•–^ production was assessed with a UV spectrophotometer at 550 nm (Hitachi U-3010, Tokyo, Japan). The percentage of superoxide inhibition of the test compound was calculated as the percentage of inhibition = {(control − resting) − (compound − resting)}/(control − resting) × 100. The software SigmaPlot was used for determining the IC_50_ values [30].

#### 3.4.3. Chemicals and Antibodies

Fluorouracil (5-FU) and bovine serum albumin (BSA) were purchased from Sigma-Aldrich (St. Louis, MO, USA). The antibodies against Bcl-2, Bax, and β-actin were purchased from Cell Signaling Technology (Danvers, MA, USA). Caspase-3 was obtained from GeneTex International Corporation (Hsinchu, Taiwan).

#### 3.4.4. Cells and Culture Medium

A549 (human lung carcinoma) and HT-29 (human colon carcinoma) cells were kindly provided by Prof. T. M. Hu and Prof. Y. Su, respectively, of National Yang-Ming University, Taipei, Taiwan.

All cell lines were cultured in Dulbecco’s modified Eagle’s medium supplemented with 10% fetal bovine serum (FBS), 100 U/mL penicillin, 100 μg/mL streptomycin, 2 μM L-glutamine, and 1 mM sodium pyruvate. The cells were incubated in an atmosphere of 37 °C and 5% CO_2_ and passaged twice a week. Cells were stored in liquid nitrogen at −155 °C. After the cells were thawed, the experiment was completed before 30 generations. The purpose was to minimize experimental errors. The compound stock solution was stored in DMSO at a concentration of 10 mM and stored at −20 °C, and finally melted immediately before use.

#### 3.4.5. Cytotoxicity Assay

A MTT colorimetric assay was used to determine cell viability. The assay was modified from that of Mosmann [32]. MTT reagent (0.5 mg/mL) was added onto the attached cells mentioned above (100 μL per 100 μL culture) and incubated at 37 °C for 3 h. Then, DMSO was added and the amount of colored formazan metabolite formed was measured by absorbance at 570 nm, using an ELISA plate reader (μ Quant). The optical density of formazan formed in control (untreated) cells was taken as 100% viability.

#### 3.4.6. Clonogenic Assay

The clonogenic assay followed as previously described with slight changes [33]. For the clonogenic assay, cells at a density of 3000 cells/well were seeded in 6-well plates for 24 h. Next, the cells were treated with compound **3** or vehicle (DMSO) and allowed to form colonies for 14 days. Colonies were washed with PBS, and the cells attached to the plastic surface were fixed in 99% MeOH for 30 min and stained with 0.2% crystal violet for 15 min. The stained cells were quantified using the ImageJ software (BioTechniques, NY, USA).

#### 3.4.7. Western Blotting Analysis

Western blot analysis was performed according to the method previously reported [34]. In brief, A549 (1 × 10^5^ cells) was seeded into 6 wells plate and grown until 85–90% confluent. Then, different concentrations (3.125, 6.25, 12.5, 25, and 50 μM) of compounds **3** and **4** were added. Cells were collected and lysed by radioimmunoprecipitation assay (RIPA) buffer. Lysates of total protein were separated by 12.5% sodium dodecyl sulfate-polyacrylamide gels and transferred to polyvinylidene difluoride (PVDF) membranes. After blocking, the membranes were incubated with anti-Bax, anti-Bcl-2 (Cell Signaling Inc., Danvers, MA, USA), anti-caspase-3, and anti-β-actin (GeneTex Inc., Irvine, CA, USA) primary antibodies at 4 °C overnight. Then, each membrane was washed with Tris-buffered saline containing 0.1% Tween 20 (TBST) and incubated with horseradish peroxidase (HRP)-conjugated secondary antibodies at room temperature, for 1 h, while shaking. Finally, each membrane was developed using an enhanced chemiluminescence (ECL) detection kit, and the images were visualized by ImageQuant LAS 4000 Mini biomolecular imager (GE Healthcare, MA, USA). The band densities were quantified using the ImageJ software (BioTechniques, NY, USA).

#### 3.4.8. Statistical Analysis

All data are expressed as mean ± SEM. Statistical analysis was carried out using Student’s t-test. A probability of 0.05 or less was considered to be statistically significant. Microsoft Excel 2019 was used for the statistical and graphical evaluation. All experiments were performed at least 3 times.

## 4. Conclusions

Three novel (**1**–**3**) and four known compounds were isolated and identified from *Penicillium citrinum*. Among the isolated compounds, compounds **2**–**5** could significantly inhibit fMLP-induced O_2_^•−^ generation, with IC_50_ values ≤ 8.28 μM. These isolated compounds are worth further research, as promising new leads for developing anti-inflammatory agents. Furthermore, compounds **3** and **4** markedly induced apoptosis of A549 cells through the mitochondrial- and caspase 3-dependent pathways (Figure 11). This suggests that compounds **3** and **4** are worth further investigation and might be expectantly developed as the candidates for the treatment or prevention of non-small cell lung cancer and liver cancer.

## Figures and Tables

**Figure 1 marinedrugs-19-00025-f001:**
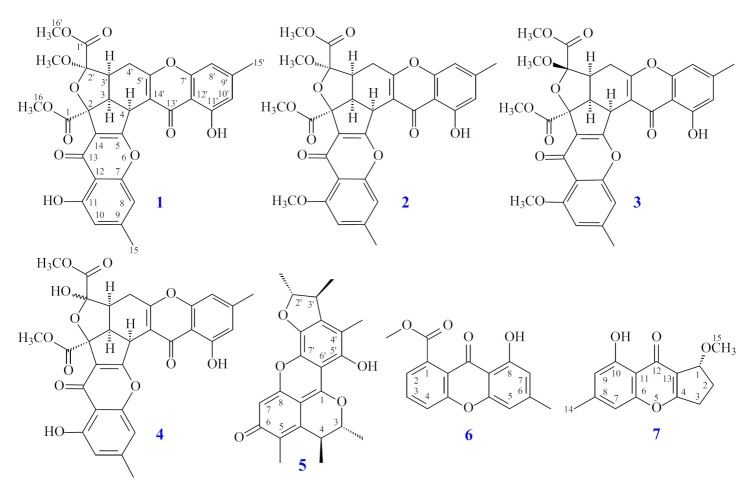
The chemical structures of compounds **1**–**7** isolated from *Penicillium citrinum*.

**Figure 2 marinedrugs-19-00025-f002:**
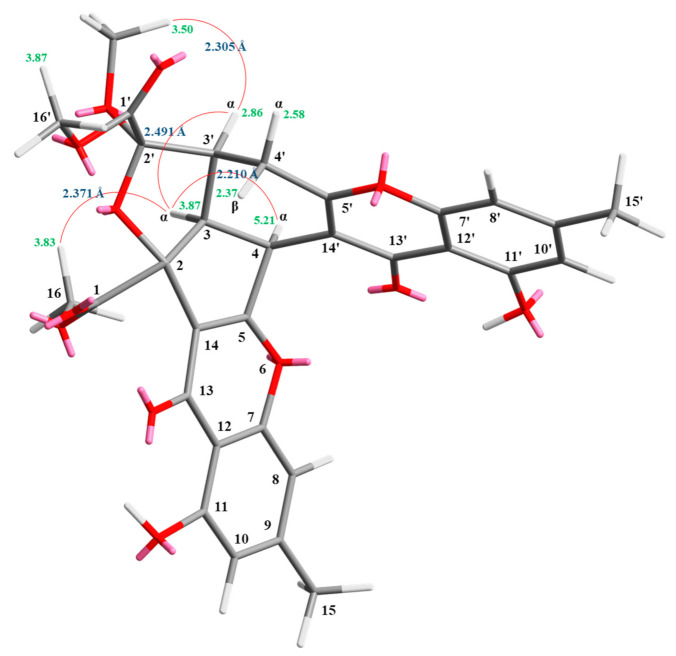
Selected ROESY correlations and relative configuration of **1**.

**Figure 3 marinedrugs-19-00025-f003:**
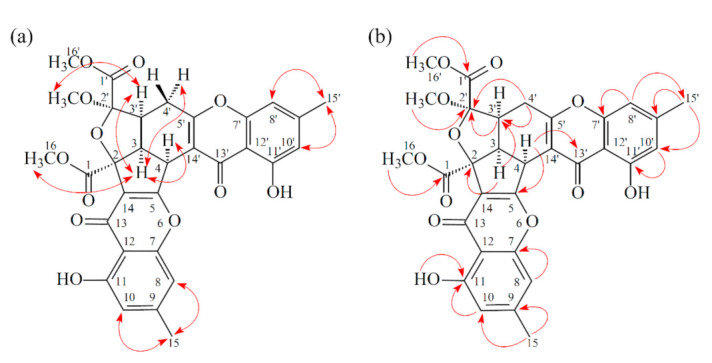
Key ROESY (**a**) and HMBC (**b**) correlations of **1**.

**Figure 4 marinedrugs-19-00025-f004:**
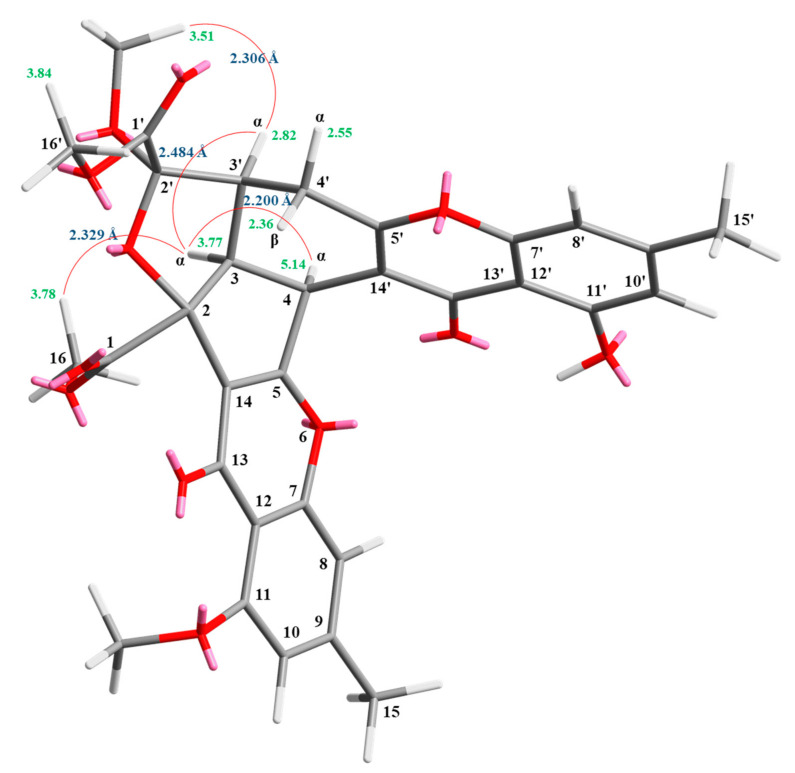
Selected ROESY correlations and relative configuration of **2**.

**Figure 5 marinedrugs-19-00025-f005:**
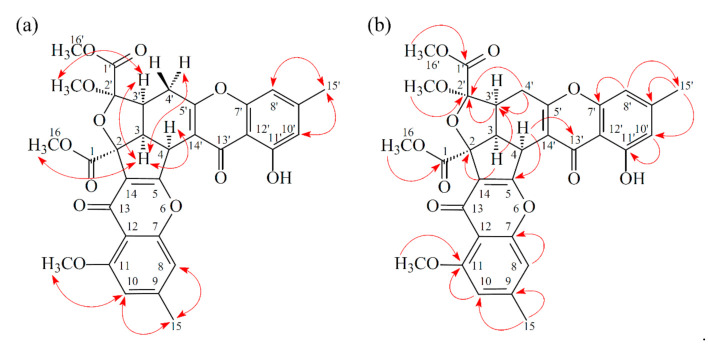
Key ROESY (**a**) and HMBC (**b**) correlations of **2**.

**Figure 6 marinedrugs-19-00025-f006:**
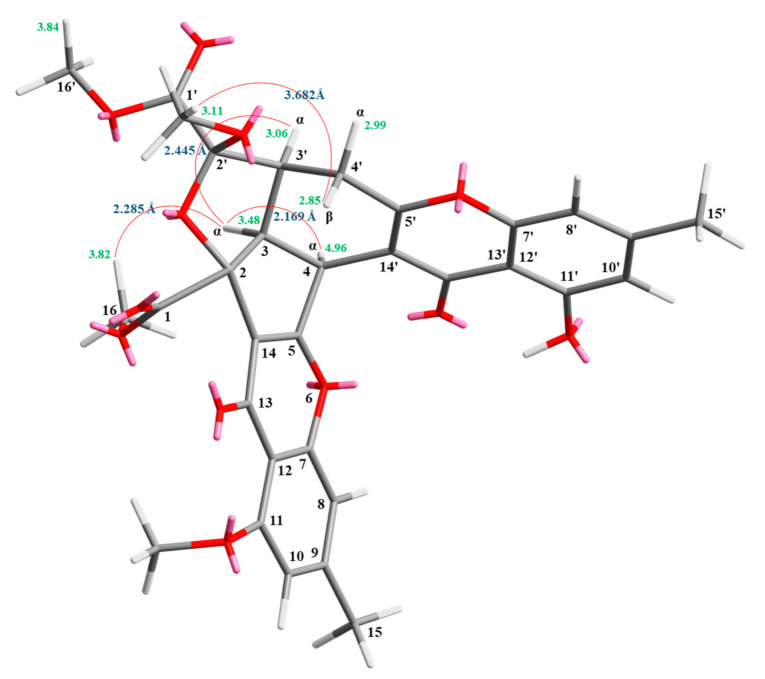
Selected ROESY correlations and the relative configuration of **3**.

**Figure 7 marinedrugs-19-00025-f007:**
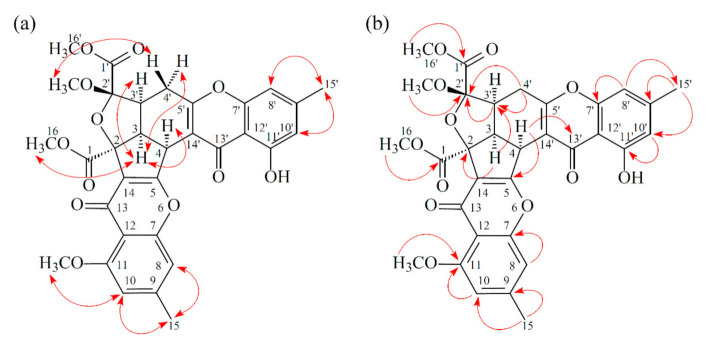
Key ROESY (**a**) and HMBC (**b**) correlations of **3**.

**Figure 8 marinedrugs-19-00025-f008:**
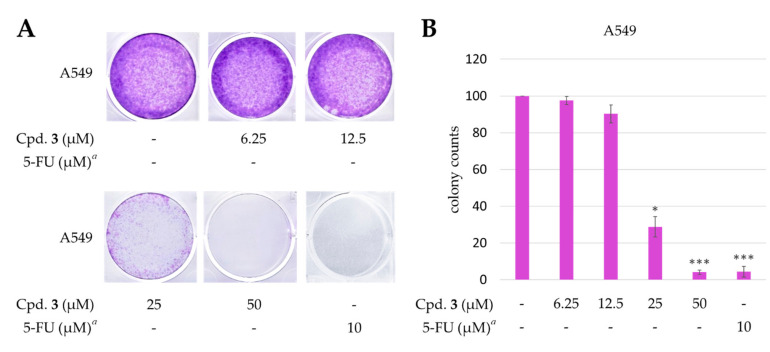
Effect of epiremisporine E (**3**) on the colony formation of A549 cells. (**A**) The effect of **3** against A549 cell colony formation. Clonogenicity was evaluated by the monolayer colony formation assay. Representative images show the blue colonies of A549 cells stained with crystal violet. (**B**) Histogram presentation of A549 cell colony quantification. ** p* < 0.05; **** p* < 0.001 compared with the control. *^a^* 5-Fluorouracil (5-FU) was used as a positive control.

**Figure 9 marinedrugs-19-00025-f009:**
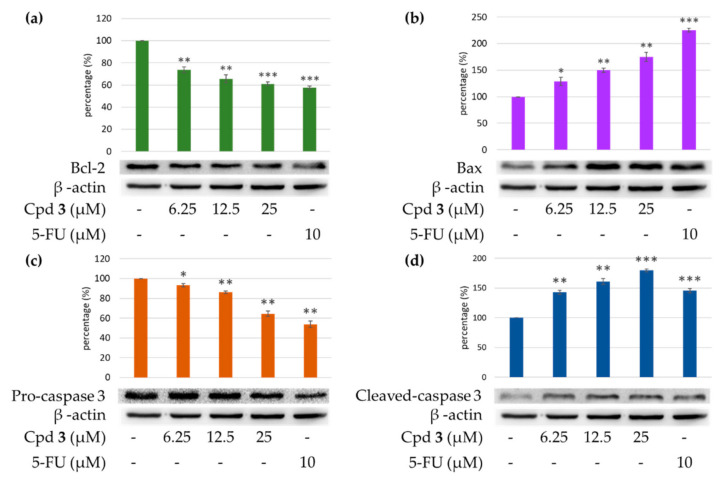
Western blot analysis for Bcl-2 (**a**), Bax (**b**), pro-caspase 3 (**c**), and cleaved-caspase 3 (**d**) in each group. Treatment with epiremisporine C (**3**) significantly reduced the expression levels of Bcl-2 and pro-caspase 3, and increased the expression levels of Bax and cleaved-caspase 3. As-terisks indicate significant differences (* *p* < 0.05, ** *p* < 0.01, and *** *p* < 0.001) compared with the control group.

**Figure 10 marinedrugs-19-00025-f010:**
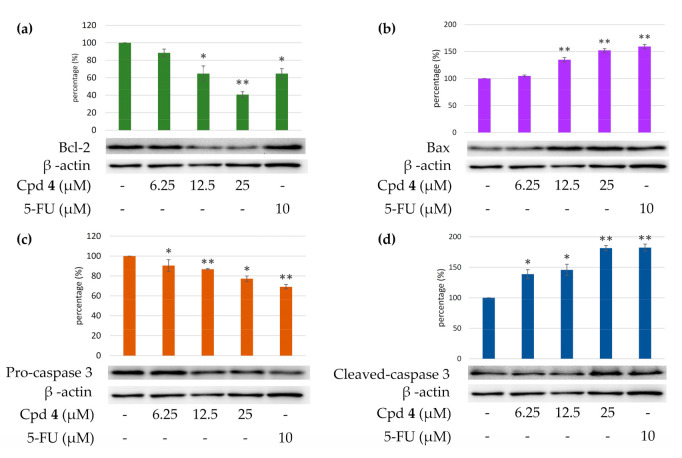
Western blot analysis for Bcl-2 (**a**), Bax (**b**), pro-caspase 3 (**c**), and cleaved-caspase 3 (**d**) in each group. Treatment with epiremisporine B (**4**) significantly reduced the expression levels of Bcl-2 and pro-caspase 3, and increased the expression levels of Bax and cleaved-caspase 3. As-terisks indicate significant differences (* *p* < 0.05 and ** *p* < 0.01) compared with the control group.

**Figure 11 marinedrugs-19-00025-f011:**
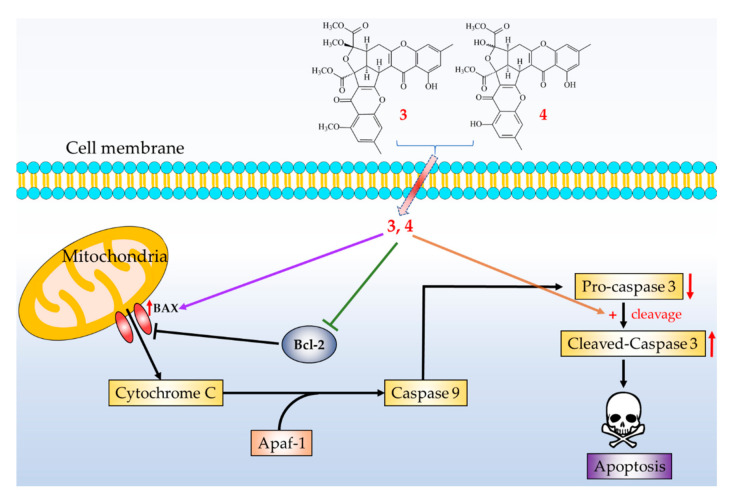
Schematic diagram for cancer cell apoptosis mechanism of compounds **3** and **4** in A549 cells.

**Table 1 marinedrugs-19-00025-t001:** The correlations between dihedral angles and vicinal coupling constants of compounds **1**–**3** and related analogues [16].

Compounds	Dihedral Angles	*J*_3′, 4′α_ (Hz)	Dihedral Angles	*J*_3′, 4′β_ (Hz)
	(H3′-C3′-C4′-H4′α)		(H3′-C3′-C4′-H4′β)	
**1** (2′*R,*3′*S*)	50.7°	5.3	169.8°	12.8
**2** (2′*R,*3′*S*)	51.1°	5.4	170.2°	12.7
**3** (2′*S,*3′*S*)	54.8°	4.7	173.9°	8.3
Epiremisporine B (2′*R,*3′*S*)	53.9°	5.9	173.5°	12.5
Epiremisporine B (2′*S,*3′*S*)	54.7°	6.4	173.8°	10.1
Epiremisporine B1 (2′*R,*3′*S*)	54.2°	6.6	173.8°	11.3
Epiremisporine B1 (2′*S,*3′*S*)	56.0°	6.5	175.2°	10.3
Remisporine B (2′*S,*3′*R*)	178.8°	12.2	61.0°	4.3

**Table 2 marinedrugs-19-00025-t002:** Inhibitory effects of compounds **1**–**7** from *Penicillium citrinum* on superoxide anion generation by human neutrophils, in response to fMLP.

Compounds	IC_50_ (μM) ^a^
Epiremisporine C (**1**)	>50
Epiremisporine D (**2**)	6.39 ± 0.40 ^e^
Epiremisporine E (**3**)	8.28 ± 0.29 ^d^
Epiremisporine B (**4**)	3.62 ± 0.61 ^e^
Penicitrinone A (**5**)	2.67 ± 0.10 ^e^
8-Hydroxy-1-methoxycarbonyl-6-methylxanthone (**6**)	>50
Isoconiochaetone C (**7**)	38.35 ± 0.21 ^c^
Ibuprofen ^b^	27.85 ± 3.56 ^c^

^a^ Concentration necessary for 50% inhibition (IC_50_). ^b^ Ibuprofen (a fMLP receptor antagonist) was used as a positive control. Results are presented as average ± SEM (*n* = 3). Values are expressed as average ± SEM (*n* = 3). ^c^
*p* < 0.05; ^d^
*p* < 0.01; ^e^
*p* < 0.001 compared with the control.

**Table 3 marinedrugs-19-00025-t003:** Cytotoxic effects of compounds **1**–**7** against A549 and HT-29 cells.

Compounds	IC_50_ (μM) ^a^	
	A549	HT-29
Epiremisporine C (**1**)	>100	>100
Epiremisporine D (**2**)	>100	>100
Epiremisporine E (**3**)	43.82 ± 6.33 ^c^	>100
Epiremisporine B (**4**)	32.29 ± 4.83 ^c^	50.88 ± 2.29 ^c^
Penicitrinone A (**5**)	49.15 ± 6.47	>100
8-Hydroxy-1-methoxycarbonyl-6-methylxanthone (**6**)	>100	>100
Isoconiochaetone C (**7**)	>100	>100
5-FU ^b^	12.52 ± 2.02 ^d^	40.92 ± 3.93 ^d^

^a^ The IC_50_ values were calculated from the slope of dose-response curves (SigmaPlot). Values are expressed as mean ± SEM (*n* = 3). ^c^
*p* < 0.05; ^d^
*p* < 0.01 compared with the control. ^b^ 5-Fluorouracil (5-FU) was used as a positive control.

**Table 4 marinedrugs-19-00025-t004:** ^1^H NMR data (500 MHz, CDCl_3_) for **1**–**3**.

Position	1	2	3
	δ_H_ (*J* in Hz)		
3	3.87 (dd, 8.9, 8.6)	3.77 (dd, 8.7, 8.6)	3.48 (dd, 10.5, 8.9)
4	5.21 (d, 8.9)	5.14 (d, 8.9)	4.96 (d, 8.9)
8	6.67 (br s)	6.76 (br s)	6.81 (br s)
10	6.62 (br s)	6.58 (br s)	6.59 (br s)
15	2.33 (s)	2.36 (s)	2.38 (s)
16	3.83 (s)	3.78 (s)	3.82 (s)
3′	2.86 (ddd, 12.8, 8.6, 5.3)	2.82 (ddd, 12.7, 8.6, 5.4)	3.06 (ddd, 10.5, 8.3, 4.7)
4′α	2.58 (dd, 15.6, 5.3)	2.55 (dd, 15.5, 5.4)	2.99 (dd, 18.6, 4.7)
4′β	2.37 (dd, 15.6, 12.8)	2.36 (dd, 15.5, 12.7)	2.85 (dd, 18.6, 8.3)
8′	6.69 (br s)	6.69 (br s)	6.69 (br s)
10′	6.70 (br s)	6.69 (br s)	6.68 (br s)
15′	2.42 (s)	2.41 (s)	2.42 (s)
16′	3.87 (s)	3.84 (s)	3.84 (s)
11-OH	12.11 (br s)	-	-
11-OMe	-	3.91 (s)	3.91 (s)
2′-OMe	3.50 (s)	3.51 (s)	3.11 (s)
11′-OH	12.30 (br s)	12.36 (s)	12.50 (s)

**Table 5 marinedrugs-19-00025-t005:** ^13^C NMR data (125 MHz, CDCl_3_) for **1**–**3**.

Position	1	2	3
	δ_C_, Type		
1	170.8, C	171.2, C	170.5, C
2	91.2, C	91.7, C	91.6, C
3	46.9, CH	46.8, CH	44.0, CH
4	36.6, CH	36.1, CH	37.3, CH
5	168.3, C	164.7, C	166.4, C
7	157.2, C	159.1, C	159.2, C
8	108.3, CH	110.7, CH	110.9, CH
9	147.4, C	145.2, C	145.1, C
10	113.1, CH	108.4, CH	108.3, CH
11	160.9, C	160.0, C	159.9, C
12	109.0, C	112.9, C	112.6, C
13	179.0, C	173.7, C	173.9, C
14	118.8, C	121.7, C	121.2, C
15	22.2, CH_3_	22.1, CH_3_	22.1, CH_3_
16	53.1, CH_3_	52.9, CH_3_	53.1, CH_3_
1′	166.7, C	167.1, C	168.5, C
2′	111.1, C	111.0, C	107.6, C
3′	48.4, CH	48.4, CH	43.3, CH
4′	27.1, CH_2_	27.1, CH_2_	25.5, CH_2_
5′	165.5, C	165.5, C	165.4, C
7′	156.0, C	156.1, C	155.9, C
8′	107.6, CH	107.5, CH	107.3, CH
9′	147.7, C	147.5, C	147.3, C
10′	112.7, CH	112.6, CH	112.2, CH
11′	160.5, C	160.5, C	160.4, C
12′	108.5, C	108.5, C	108.3, C
13′	179.7, C	179.8, C	180.6, C
14′	112.8, C	113.0, C	111.7, C
15′	22.4, CH_3_	22.4, CH_3_	22.4, CH_3_
16′	52.8, CH_3_	52.6, CH_3_	52.8, CH_3_
11-OMe	-	56.3, CH_3_	56.4, CH_3_
2′-OMe	52.7, CH_3_	52.7, CH_3_	52.2, CH_3_

## Data Availability

The data presented in this study are available in the main text and the Appendix A of this article.

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
