# Peer review of "Rare Chromone Derivatives from the Marine-Derived Penicillium citrinum with Anti-Cancer and Anti-Inflammatory Activities"

_marinedrugs, 2021, doi:10.3390/md19010025_

Round 1
Reviewer 1 Report
Manuscript 1022601 describes the isolation, structure elucidation and bioactivity evaluation of chromone derivatives from a marine-derived fungus, Penicillium citrinum. Whilst the data seem reliable and the conclusions appropriate, unfortunately the manuscript must undergo extensive editing and revision before it can be accepted for publication to address significant and frequent deficiencies in the correct use of English language.
Author Response
Please see an attached file.

Reviewer 2 Report
please see attached file

Author Response
Please see an attached file.

Reviewer 3 Report
Manuscript: «Rare Chromone Derivatives from the Marine-Derived Penicillium citrinum with Anti-Cancer and Anti-Inflammatory Activities» by Chu et al.
The manuscript describes the isolation and chemical characterisation of three new and four known compounds from the fungi P. citrinum. All four compounds were tested for cytotoxic activity against cancer cell lines and and ability to suppress generation of superoxide anion. Two of the new compounds were also tested for their ability to affect apoptosis in A549 cells.
General comments:
The manuscript is in general scientifically sound and well written.
Some info should be included in the introduction about the background of the isolate.
There are in general some duplication in the reporting of data: For the structures, key values for proton and carbon spectra are listed in both the results section (as text) and in the experimental section (as tables). Identical MS and IR data are also reported in both these two sections (i.e. ‘results’ and ‘experimental’). This should be avoided.
Structural data for the known compounds are missing. Key data should be reported such as HR-MS data, isolation protocol and amounts of isolated compounds (missing for 5-7). Data should also be included in SI.
NMR spectra in the SI are not annotated.
Why was just compound 3 tested in the clonogenic assay when compounds 4 and 5 also were active in the NTT assay? This should be explained.
Avoid using the term ‘isolates’ about isolated compounds, as this term is easily confused with isolated microorganisms.
There is no discussion of the observed differences in bioactivities between the studied compounds, even though some of them are very similar. This must be included.
Specific comments
L28 ‘inhibitory activities’ needs to be specified. What is inhibited?
L35 What is meant by ‘special structures’? This must be put in some context.
L45 What is ment by the statement ‘cancer is the top ten cause of death’?
L48+49 What is meant by ‘this species’? Please specify.
L50 ‘figus’, please correct.
L56 The start of this section is rather abrupt. Please start with a brief description of the fermentation and extraction.
L59 Please reorder the structures in figure 1 so they follow a horizontal pattern
L59 Why are the protons on carbon 4’ drawn for compound 3?
L60 Write ‘hydroxyl’ instead of ‘OH’ (same for L98 and 133)
L124 Write ‘Figure 5a’
L125 Write ‘Figure 5b’
L165 Include more information on the known compounds
L166 Write ‘isolated compounds’ instead of ‘isolates’
L173 Write ‘evaluating by their ability to suppress’
L183 Tables 1 and 2 need to be changed to improve readability (alignment of lines)
L189 Replace ‘were’ with ‘are’
L190 Please specify what is inhibited (a cell is not inhibited, but the growth, division or metabolism of a cell can be inhibited)
L188-193 All the data from table 2 is duplicated in the text
L198 Selectivity is not demonstrated when only one cell line is tested
L203 Size of photos should be increased
L248-9 What is the unit of the mesh size?
L258 I do not understand the meaning of the statement ‘P. citrinium was used throughout the research process’. Please explain.
L276 What is the unit of the mesh size?
L386 It is not necessary to repeat the specific values one more time.
L406Where does figure 11 belong? I cannot find any reference to it in the text.
Author Response
Please see an attached file.

Reviewer 4 Report
To determine the relative stereochemistry, the authors used mainly ROESY data with MM2 calculation. It should be mentioned that J values are important information for discussion of cyclic portion such as 6-membered ring system and also 5-membered ring. Relative large coupling constant values 3JH3,H4, and 3JH3,H3’ and 3JH3’,H-4’beta should be re-considered carefully. In ROESY spectrum, some vicinal J coupling will give some correlations like COSY, so phase and shape of cross peaks are important. Please add expanded ROESY spectra for Supporting Information.
Author Response
Please see an attached file.

Round 2
Reviewer 2 Report
I cannot see Figure 11 in the pdf version that I downloaded, I can only see the Figure caption. Overall, the manuscript has been revised as my suggestion.
Author Response
Please see an attached file.

Reviewer 3 Report
Revised manuscript: «Rare Chromone Derivatives from the Marine-Derived Penicillium citrinum with Anti-Cancer and Anti-Inflammatory Activities» by Chu et al.
General comments:
The manuscript is improved, but there are some points that should be revisited by the authors.
There are still duplication in the reporting of data: For the structures, key values for proton and carbon spectra are listed in both the results section (as text) and in the experimental section (as tables). Identical MS and IR data are also reported in both these two sections (i.e. ‘results’ and ‘experimental’). This has not been corrected.
The structure elucidation of 3 focus on the differences between 3 and 1, more emphasis should be given to the difference between 3 and 2.
NMR spectra in the SI are still not annotated.
Why was just compound 3 tested in the clonogenic assay when compounds 4 and 5 also were active in the NTT assay? This should be explained, not to me as a referee, but to the readers in the manuscript.
The discussion of the observed differences in bioactivities between the studied compounds must be expanded. For instance, how do you explain the difference between 2 and 3?
I cannot see figure 11 in my version of the manuscript.
Author Response
Please see an attached file.

Reviewer 4 Report
The authors rechecked J values with related compounds. However dihedral angles between H-3’ and H-4’a, and between H-3’ and H-4’bare seems to be ca 30 degrees and ca 90 degrees based on the 3D structures by Chem3D. It is not match with J values for 3JH-3’,H-4’alpha and 3JH-3’,H-4’beta, especially 12.8Hz, 12.7Hz, and 8.3Hz for 3JH-3’,H-4’beta of compounds 1, 2, and 3, respectively. This discrepancy is severe problem on the structure determination of compounds 1 -3, together with previously reported compounds (Mar. Drugs 2015, 13, 5219). Please show clear explanation and list up dihedral angles for 3D structures.
Author Response
Please see an attached file.

Round 3
Reviewer 4 Report
In the revised version, selected vicinal coupling constant values and calculated dihedral angles are summarized in Table 1. Based on Karplus equation 3J values for dihedral angles between 60 to 90 degrees are less than 3Hz, but 3JH3’,H4’beta are very large values between 8.3Hz for compound 3 and 12.8Hz for compound 1.
This discrepancy strongly indicates that structural determination should be re investigated including previously reported epiremisporine B analogues.
Author Response
Please see an attached file.
